# Testis-specific serine/threonine kinase dTSSK2 regulates sperm motility and male fertility in *Drosophila*
Ju Peng[1], Angyang Sun[1], Jie Zheng[1], Na Zhang[1], Xuedi Zhang [2]✉ & Guanjun Gao [1]✉

Serine/threonine kinases of the TSSK (Testis-Specific Serine/Threonine Kinase) family play crucial roles in spermatogenesis and male fertility across species, but the underlying regulatory mechanism remains incompletely understood. In this study, we identified and characterized a novel TSSK homolog in *Drosophila*, named dTSSK2 (CG9222), which functions as the ortholog of human TSSK4. dTSSK2 is specifically expressed in the testis and localizes to individualization complexes during spermiogenesis. Disruption of dTSSK2 severely compromises sperm motility, leading to failed sperm transit into the seminal vesicle and male infertility. Phosphoproteomic analyses reveal that dTSSK2 coordinates sperm flagella assembly and motility by phosphorylating proteins involved in microtubule organization, organelle assembly, and flagella structure. Notably, dTSSK2 phosphorylates the substrate Gudu at Ser9, which partially contributes to individualization complex integrity and sperm motility. These findings elucidate the critical role of dTSSK2-mediated phosphorylation in regulating *Drosophila* male fertility.

Spermatogenesis is a highly dynamic and evolutionarily conserved process that involves proliferation and differentiation of spermatogonia into round spermatocytes, followed by their transformation into elongated, mature sperm cells[1–3]. The terminal phase of this process, spermatogenesis, is characterized by conserved morphological and biochemical transitions, such as acrosome formation, flagellar development, cytoplasmic extrusion, and chromatin condensation[4–6]. These events are tightly regulated, particularly during spermiogenesis, where post-transcriptional and post-translational modifications (PTMs) orchestrate the maturation of sperm cells due to transcriptional silencing in haploid cells[7]. Among the various PTMs known to play critical roles in spermatogenesis, the specific contributions of phosphorylation in regulating these processes remain intensively underexplored.

Testis-Specific Serine/Threonine Kinases (TSSKs) family represents an evolutionarily conserved family of proteins that are exclusively expressed in the testis and are indispensable for sperm maturation and male fertility. The mammalian genome encodes six TSSK isoforms, and their functional ablation has been linked to infertility through defective sperm development and motility[8]. For example, TSSK1 and TSSK2 are required for the chromatoid body function and sperm tail maturation, while TSSK4 is essential for maintaining the structural integrity of flagellar axonemes[9–12]. These findings have also spurred interest in TSSKs as potential drug targets for male contraceptive development, as pharmacological inhibition of TSSKs could disrupt sperm maturation. While in vitro studies have identified

phosphorylation substrates of mammalian TSSKs, such as TSKS, ODF2, and CREB, the mechanistic roles of TSSK-mediated phosphorylation in sperm maturation remain poorly understood[8,13,14]. Furthermore, the redundancy among mammalian TSSK isoforms complicates efforts to pinpoint specific functions for each kinase.

The conservation of spermatogenesis across species underlines an opportunity to leverage simpler model organisms to study fundamental aspects of sperm maturation. In *Drosophila melanogaster*, spermiogenesis shares striking similarities with mammals, including conserved pathways of cytoskeletal remodeling and chromatin condensation, as well as the involvement of TSSK orthologues[15]. Unlike mammals, *Drosophila* has only two TSSK homologs, simplifying the study of their specific function. For example, the recently characterized *Drosophila* TSSK (dTSSK, encoded by CG14305) regulates sperm chromatin remodeling by phosphorylating substrates such as the protamine-like protein Mst77F and the transition protein Mst33A, which are essential for histone-to-protamine transition[16]. However, the exact roles of other *Drosophila* TSSKs in sperm maturation, particularly in the regulation of sperm motility, remain unknown.

In this study, we address two major gaps in our understanding of TSSKs: how TSSK-mediated phosphorylation governs sperm motility and whether TSSK functions are conserved across species. Using *Drosophila* as a model, we demonstrate that dTSSK2 (encoded by CG9222) is essential for sperm flagellar motility and male fertility. Through CRISPR/Cas9 mutagenesis,

[1]School of Life Science and Technology, ShanghaiTech University, Shanghai, China. [2]School of Basic Medical Sciences, Suzhou Medical College, Soochow University, Suzhou, Jiangsu Province, China. ✉e-mail: xdzhang11@suda.edu.cn; gaogj@shanghaitech.edu.cn

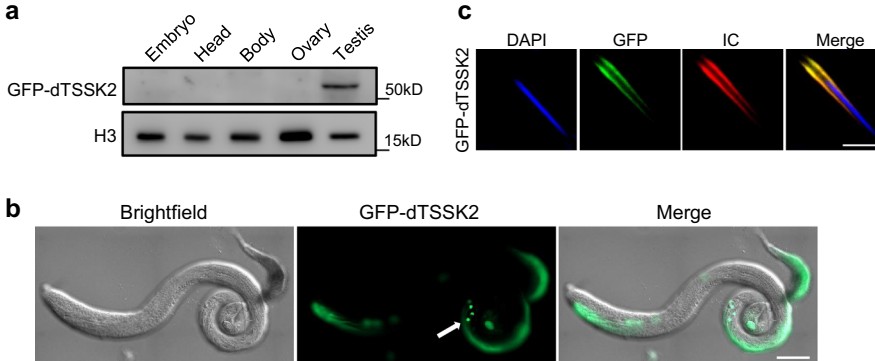

**Fig. 1 | Localization of *Drosophila* CG9222 in testicular tissue. a** WB analysis showing the specific expression of the dTSSK2 protein in testicular tissue. Different tissues (including embryo, head, body, ovary, and testis) were dissected from dTSSK2 transgenetic flies tagged with GFP and driven by its endogenous promoter. Tissue homogenates were used for WB analysis against anti-GFP, and the observed band size of GFP-dTSSK2 is consistent with its predicted size. H3 was used as a loading control. **b** Live imaging showing the distribution of dTSSK2 protein in testes of GFP-dTSSK2 transgenic flies (GFP-dTSSK2, green). Sperm bundles are indicated by the white arrow. Scale bar, 100 µm. **c** Live imaging showing the distribution of dTSSK2protein of IC phase during spermiogenesis (sperm DNA stained with DAPI, blue; GFP-dTSSK2, green; ICs stained with phalloidin, red). Scale bar, 5 µm.

phosphoproteomic analysis, and functional rescue experiments, we identified Gudu—a key component of the individualization complex (IC)—as a direct substrate of dTSSK2. Phosphorylation of Gudu at Serine 9 partially contributes to sperm motility, suggesting that dTSSK2 regulates cytoskeletal remodeling during spermiogenesis. Intriguingly, human TSSK4 can fully rescue fertility in dTSSK2 mutant flies, underscoring the evolutionary conservation of TSSK functions. By elucidating the role of dTSSK2 in sperm motility, our findings not only advance the understanding of TSSK-mediated phosphorylation in spermatogenesis but also provide a foundation for translational research into human reproductive health. These results contribute to the development of therapeutics for male infertility and inform strategies for male contraceptive development.

## Results

### *Drosophila* CG9222 is highly enriched in testis tissue

Our previous work identified the *Drosophila* CG14305/dTSSK as a key regulator of sperm chromatin condensation during spermiogenesis[16]. However, the *Drosophila* genome likely encodes additional TSSK family members that contribute to the regulation of normal sperm maturation. Utilizing publicly available datasets from modENCODE RNA-sequencing and developmental proteomic studies accessible via FlyBase (www.flybase.org)[17–19], we identified another gene, CG9222, as a direct ortholog of human TSSKs.

To determine the conservation of CG9222 with known TSSKs, we conducted a comparative analysis of its amino acid sequence against the five human TSSKs[20] (Fig. S1a). Multiple sequence alignments revealed that CG9222 shares 29–33% sequence identity and 42–52% similarity with human TSSKs (Fig. S1b). Phylogenetic analysis categorized CG9222 within the human TSSK4 subgroup (Fig. S1c), suggesting a close evolutionary relationship to TSSK4. Sequence alignment also confirmed that CG9222 retains key kinase features, including an ATP-binding region (105–127 aa), a predicted protein kinase active site (195–207 aa), and a putative activation loop phosphorylation motif (236–238 aa). Structural modeling using AlphaFold Protein Structure Database revealed that the S_TKc kinase catalytic domain of CG9222 (76–333 aa) shares a high degree of similarity with the catalytic domains of human TSSKs[21] (Fig. S2). Given the sequence homology and structural similarity, we hereinafter refer to CG9222 as dTSSK2.

To investigate the biological role of dTSSK2 in *Drosophila* spermiogenesis, we initially examined its tissue-specific expression and subcellular localization. For this purpose, western blot (WB) analysis was performed on transgenic flies expressing dTSSK2 tagged with Flag-GFP under its native promoter. The results demonstrated that dTSSK2 was highly expressed in testicular tissue (Figs. 1a and S3a). Whole-mount imaging of testes further revealed that dTSSK2 localizes to the flagella of developing spermatids, with the strongest fluorescence signals observed in late-stage testicular tissue (Figs. 1b and S3b).

High-resolution imaging of dissected testicular tissues shows that these fluorescent signals were concentrated on both sides of the nuclei in needle-stage spermatids, coinciding with the location of the IC (Fig. S3c).

To further confirm the association of dTSSK2 with IC, we performed TRITC phalloidin staining on testicular tissue lysates to visualize actin structures, which are hallmarks of the IC. The overlay of phalloidin staining and dTSSK2 fluorescence revealed a nearly identical distribution pattern, indicating that dTSSK2 colocalizes with the IC (Figs. 1c and S3d). These observations strongly suggest that dTSSK2 plays a critical role in the individualization and maturation processes during spermiogenesis

### Disruption of dTSSK2 severely impairs male fertility

To investigate the functional role of dTSSK2 in spermiogenesis, we employed CRISPR/Cas9 technology to generate a yielded dTSSK2-knockout (dTSSK2$^{-/-}$) strain. Co-injection of Cas9 mRNA and dTSSK2-specific guide RNAs (gRNAs) into *Drosophila* embryos, followed by germline transformation, resulted in flies carrying a premature stop codon that eliminated the S_TKc kinase catalytic domain of dTSSK2 (Fig. 2a). Genomic PCR sequencing confirmed successful mutagenesis of the dTSSK2 locus (Fig. 2a). Quantitative PCR (qPCR) analysis further demonstrated a significant reduction in dTSSK2 transcript levels in dTSSK2$^{-/-}$ flies compared to wild-type controls (Fig. 2b).

While adult dTSSK2$^{-/-}$ flies exhibited normal external morphology, male dTSSK2$^{-/-}$ flies exhibited severe fertility defects (Fig. 2c). To investigate potential abnormalities in sperm localization, we generated transgenic flies expressing GFP-tagged protamine Mst35Bb, a marker for mature sperm[22]. In wild-type flies, GFP-labeled sperm were predominantly located in seminal vesical. However, in age-matched dTSSK2$^{-/-}$ flies, mature sperm accumulated within the testis and failed to completely migrate into the seminal vesicle (Fig. 2d), suggesting that impaired sperm transport may underlie the fertility defects in dTSSK2$^{-/-}$ males.

To assess whether dTSSK2 deletion affects sperm morphology or the structure of the IC during spermiogenesis, we performed detailed imaging analyses of dTSSK2$^{-/-}$ testes. No significant abnormalities were observed in the structure or localization of the IC (Fig. S4a, b), nor were there visible defects in sperm flagella morphology under bright-field microscopy (Fig. S4c). These findings indicate that the loss of dTSSK2 does not disrupt sperm structure or individualization.

To uncover the cause of impaired sperm transport, we assessed sperm motility in dTSSK2$^{-/-}$ flies. Our analysis revealed a marked reduction in sperm motility compared to wild-type controls (Fig. 2e). These findings strongly indicate that the absence of dTSSK2 compromises male *Drosophila* fertility primarily by impacting sperm flagellar motility, rather than affecting sperm morphology or individualization during spermiogenesis.

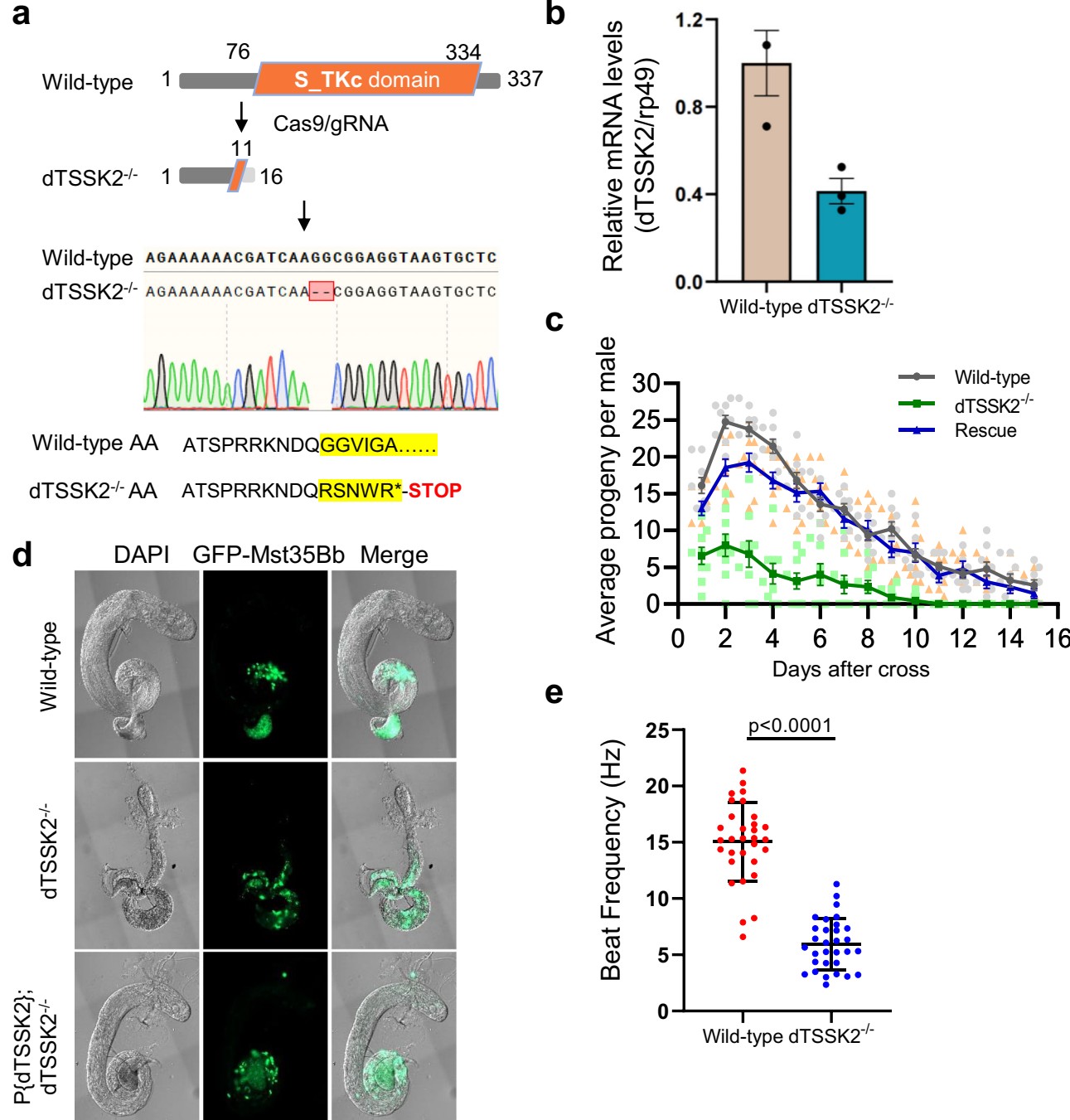

**Fig. 2 | Phenotypic defects produced by the *Drosophila* CG9222 mutation.**
**a** CRISPR/Cas9-mediated dTSSK2 gene disruption in *Drosophila*. Sequencing results revealed the deletion of two bases in the coding region of the dTSSK2 gene, resulting in formation of a premature stop codon and complete removal of the S_TKc kinase catalytic domain. **b** The mRNA expression levels in testicular extracts from wild-type and dTSSK2$^{-/-}$ flies were assessed by RT-qPCR analysis. **c** Qualitative fertility assay of wild-type (w$^{1118}$), dTSSK2$^{-/-}$ and P{dTSSK2};

dTSSK2$^{-/-}$ male flies ($n = 10$ per group). Data are mean ± SEM. **d** Live imaging of testes and seminal vesicles of the indicated genotypes. Sperm nuclei (green) are labeled with protamine GFP-Mst35Bb. Cytological examination showing defects of whole testes in dTSSK2$^{-/-}$ male flies. Scale bar, 100 μm. **e** Quantification of tail-beat frequency of sperm by wild-type (w$^{1118}$) or dTSSK2 mutant males ($n = 30$ per group). Data are mean ± SEM.

## The K105 and T237 sites of dTSSK2 are essential for sperm flagella motility

To evaluate whether the kinase activity of dTSSK2 is critical for its in vivo function, we generated transgenic flies expressing mutations in key conserved sites of dTSSK2 (Flag-dTSSK$^{K105M}$ and GFP-dTSSK$^{K105M}$) in which the conserved lysine residue at position 105 in the ATP-binding site of the S_TKc kinase domain was mutated to methionine[14,23,24] (Fig. S1a). Western

blot (WB) analysis revealed a substantial reduction in the expression levels of the Flag-tagged dTSSK$^{K105M}$ protein compared to the wild-type version (Fig. 3a). Immunofluorescence microscopy using GFP-tagged dTSSK$^{K105M}$ as a tracer revealed markedly reduced co-localization of the mutant protein with the IC during the later stages of spermiogenesis (Fig. S5a, b).

Phenotypic analysis of the dTSSK$^{K105M}$ mutant flies revealed significantly impairments in male fertility (Fig. 3b). GFP-tagged protamine

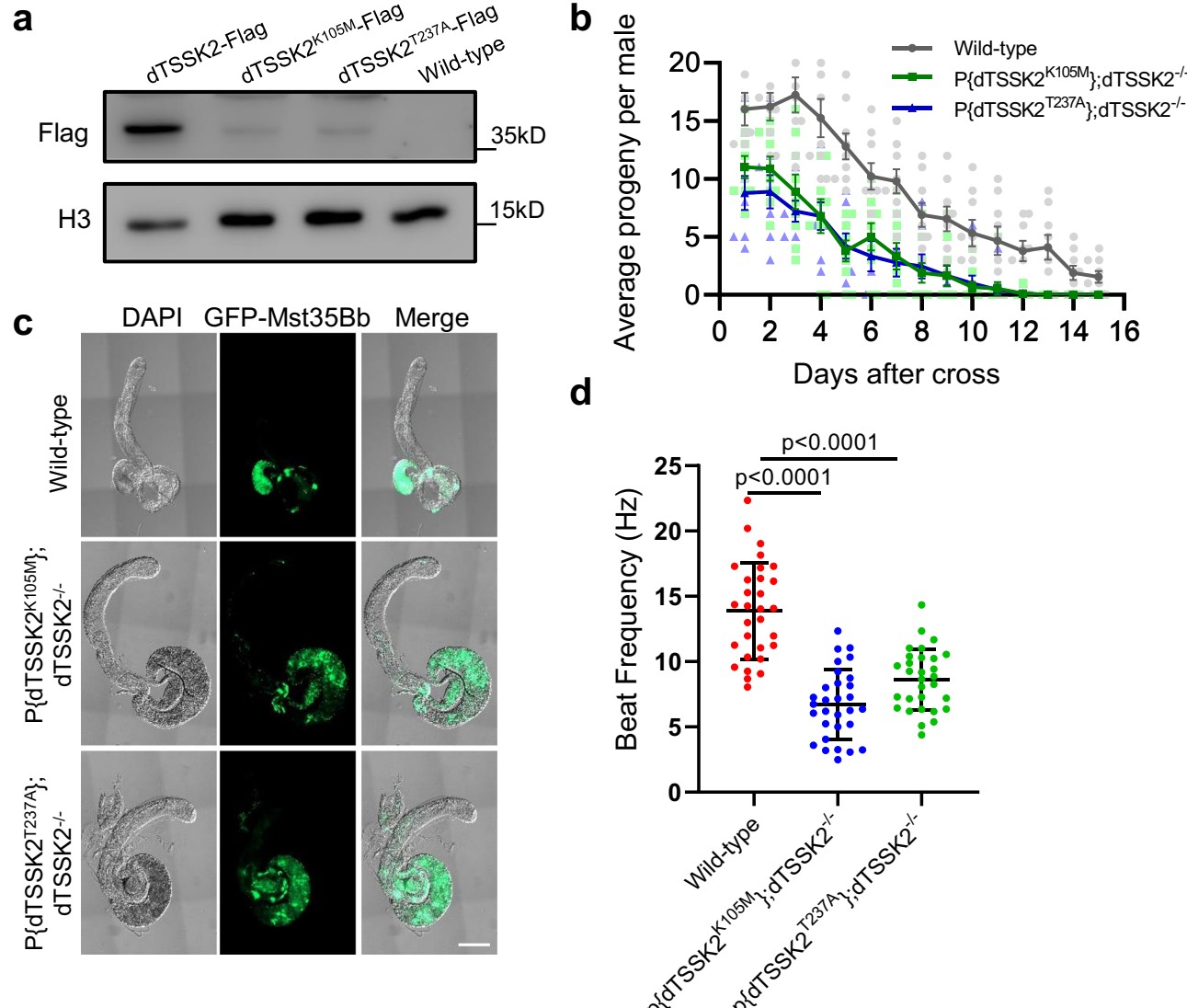

**Fig. 3 | The K105 and T237 sites of dTSSK2 are essential for sperm motility. a** WB analysis showing expression in testicular of Flag-dTSSK2, Flag-dTSSK2$^{K105M}$, Flag-dTSSK2$^{T237A}$ and wild-type flies. H3 was used as a loading control. **b** Qualitative fertility assay of wild-type (w$^{1118}$), P{dTSSK2$^{K105M}$}; dTSSK2$^{-/-}$ and P{dTSSK2$^{T237A}$}; dTSSK2$^{-/-}$ male flies (n = 10 per group). Data are mean ± SEM. **c** Live imaging of testes and seminal vesicles of the indicated genotypes. Sperm nuclei (green) are labeled with protamine GFP-Mst35Bb. Cytological examination showing defects of whole testes in dTSSK2$^{-/-}$ male flies. Scale bar, 100 μm. **d** Quantification of tail-beat frequency of sperm by wild-type (w$^{1118}$), P{dTSSK2$^{K105M}$}; dTSSK2$^{-/-}$ and P{dTSSK2$^{T237A}$}; dTSSK2$^{-/-}$ males (n = 30 per group). Data are mean ± SEM.

Mst35Bb was used to monitor the localization of mature sperm, and the results demonstrated that the conserved site mutation inhibited the efficient entry of mature sperm into the seminal vesicle, closely mirroring the phenotype of dTSSK2$^{-/-}$ flies (Fig. 3c). Sperm motility assays confirmed that the mutation caused a significant reduction in motility (Fig. 3d), providing a plausible explanation for the impaired sperm transport and reduced fertility. We also generated transgenic flies expressing another mutation in dTSSK2, dTSSK2$^{T237A}$, which disrupted the conserved potential activation loop phosphorylation motif[24]. Similar to dTSSK2$^{-/-}$ and dTSSK2$^{K105M}$ flies, dTSSK2$^{T237A}$ mutants exhibited reduced fertility, impaired sperm transport into seminal vesicle, and diminished sperm motility (Figs. 3a–d and S5a, b).

Interestingly, WB analysis showed that the expression levels of both mutant proteins (dTSSK$^{K105M}$ and dTSSK2$^{T237A}$) were significantly reduced compared to the wild-type protein, and immunofluorescence analysis revealed decreased localization to the IC. These results suggest that mutations in these conserved sites not only disrupt kinase activity but also compromise protein stability (or translational efficiency), collectively contributing to the observed defects in sperm motility and fertility.

## Human TSSK4 rescues the defective motility of sperm in dTSSK2$^{-/-}$ spermatids

To further investigate the functional conservation of TSSKs in spermiogenesis across various species and determine whether human TSSKs can rescue the defective phenotype of dTSSK2$^{-/-}$ flies. We generated transgenic flies expressing human TSSK4 under the control of the endogenous dTSSK2 promoter. Human TSSK4 was chosen because it exhibits a close evolutionary relationship, as shown by previous phylogenetic analyses (Fig. S1b, c). After introducing the human TSSK4 transgene into the dTSSK2$^{-/-}$ background, we observed that the human TSSK4-rescued flies exhibited a significant restoration of male fertility, which was otherwise severely impaired in dTSSK2$^{-/-}$ flies (Fig. 4a). This suggests that human TSSK4 can functionally replace dTSSK2 in supporting male fertility in *Drosophila*. Next, we examined the morphology and localization of protamine Mst35Bb in the testes of human TSSK4-rescued flies to assess whether TSSK4 could rectify the phenotypic defects induced by the loss of dTSSK2. Our analysis revealed that human TSSK4 effectively restored the proper localization of mature sperm, reversing the aberrant distribution observed in dTSSK2$^{-/-}$ flies (Fig. 4b). Furthermore, we evaluated the motility of sperm flagella in human TSSK4-

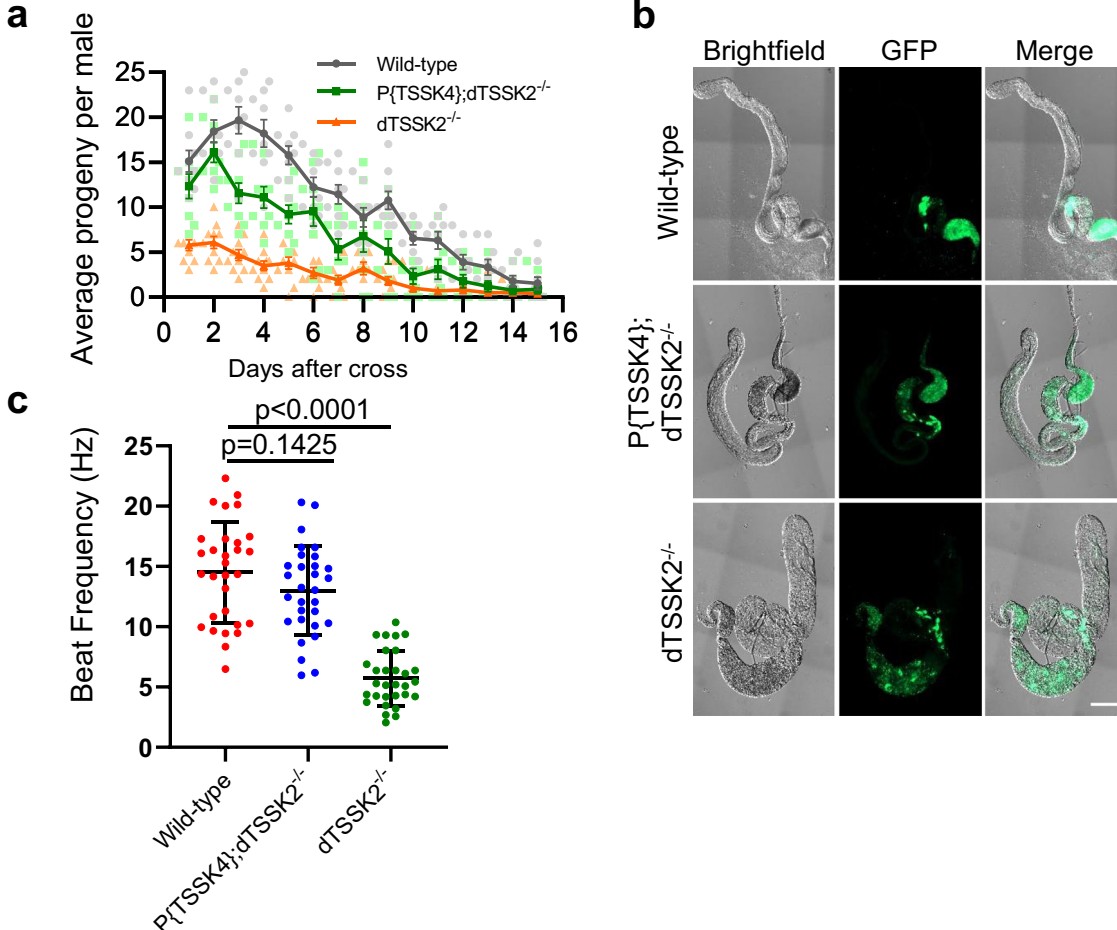

**Fig. 4 | Human TSSK4 can rescue the major phenotypic defects caused by dTSSK2 deletion. a** Qualitative fertility assay of wild-type (w[1118]), P{TSSK4}; dTSSK2[−/−] and P{dTSSK2}; dTSSK2[−/−] male flies (*n* = 10 per group). Data are mean ± SEM. **b** Live imaging of testes and seminal vesicles of the indicated genotypes. Sperm nuclei (green) are labeled with protamine GFP-Mst35Bb. Cytological examination showing defects of whole testes in dTSSK2[−/−] male flies. Scale bar, 100 μm. **c** Quantification of tail-beat frequency of sperm by wild-type (w[1118]), P{TSSK4}; dTSSK2[−/−] and P{dTSSK2}; dTSSK2[−/−] male flies (*n* = 30 per group). Data are mean ± SEM.

rescued flies. Consistent with the restoration of fertility and sperm localization, human TSSK4 substantially rescued the defective sperm flagellar motility caused by dTSSK2 deficiency (Fig. 4c). Taken together, these results demonstrate that dTSSK2 plays a critical role in the maturation and motility of spermatids during *Drosophila* spermiogenesis. Moreover, the ability of human TSSK4 to rescue the dTSSK2[−/−] phenotype highlights the evolutionary conservation of TSSK function between *Drosophila* and humans, underscoring the fundamental role of TSSKs in regulating spermiogenesis across species.

### Phosphoproteomic screening identifies potential physiological substrates of dTSSK2

Given the essential role of dTSSK2 kinase activity in spermiogenesis, we sought to identify its potential phosphorylated substrates by performing a comparative phosphoproteomic analysis of testes from wild-type (w[1118]) and dTSSK2[−/−] male flies. To achieve this, we used proteomic and phosphoproteomic approaches to investigate differentially phosphorylated sites in the two genotypes. For the phosphoproteomic analysis, testes from three biological replicates of dTSSK2[−/−] and wild-type flies were dissected and processed. Phosphopeptides were enriched and analyzed using liquid chromatography-tandem mass spectrometry (LC-MS/MS) on a high-resolution mass spectrometer to comprehensively profile the phosphoproteome. This analysis identified 143 phosphopeptides that were significantly differentially phosphorylated in dTSSK2[−/−] testes compared to wild-type testes (fold-change > 1.5-fold, *P* < 0.05). Among these, 81 phosphopeptides

exhibited decreased phosphorylation, while 62 showed increased phosphorylation (Fig. 5a and Supplementary Table 2). To gain deep insight into the biological processes potentially regulated by dTSSK2-mediated phosphorylation, we performed Gene Ontology (GO) analysis on the proteins corresponding to the phosphopeptides with reduced phosphorylation in dTSSK2[−/−] testes. This analysis revealed significant enrichment of pathways associated with critical spermiogenesis processes, including microtubule cytoskeleton organization, supramolecular fiber organization, male gamete generation, and actin filament-based (Fig. 5b). These results align with the subcellular localization of dTSSK and the morphological defects observed in dTSSK2[−/−] spermatids. Collectively, these data suggest that dTSSK2-mediated phosphorylation plays a pivotal role in regulating multiple spermiogenesis pathways.

### Gudu is a potential substrate of dTSSK2 involved in regulating sperm motility

The *Drosophila* protein Gudu is known to play a critical role in spermiogenesis, as its absence leads to defective IC formation and male infertility[25]. Our phosphoproteomic analyses have identified two primary phosphorylation sites within a Gudu phosphopeptide (Supplementary Tables 1 and 2). Among these, phosphorylation at Ser9 was significantly reduced in dTSSK2[−/−] testes compared to wild-type controls (Fig. 5c). Notably, proteomic profiling revealed no changes in the overall expression levels of the Gudu protein in dTSSK2[−/−] flies (Supplementary Table 2), suggesting that dTSSK2 specifically regulates Gudu phosphorylation rather

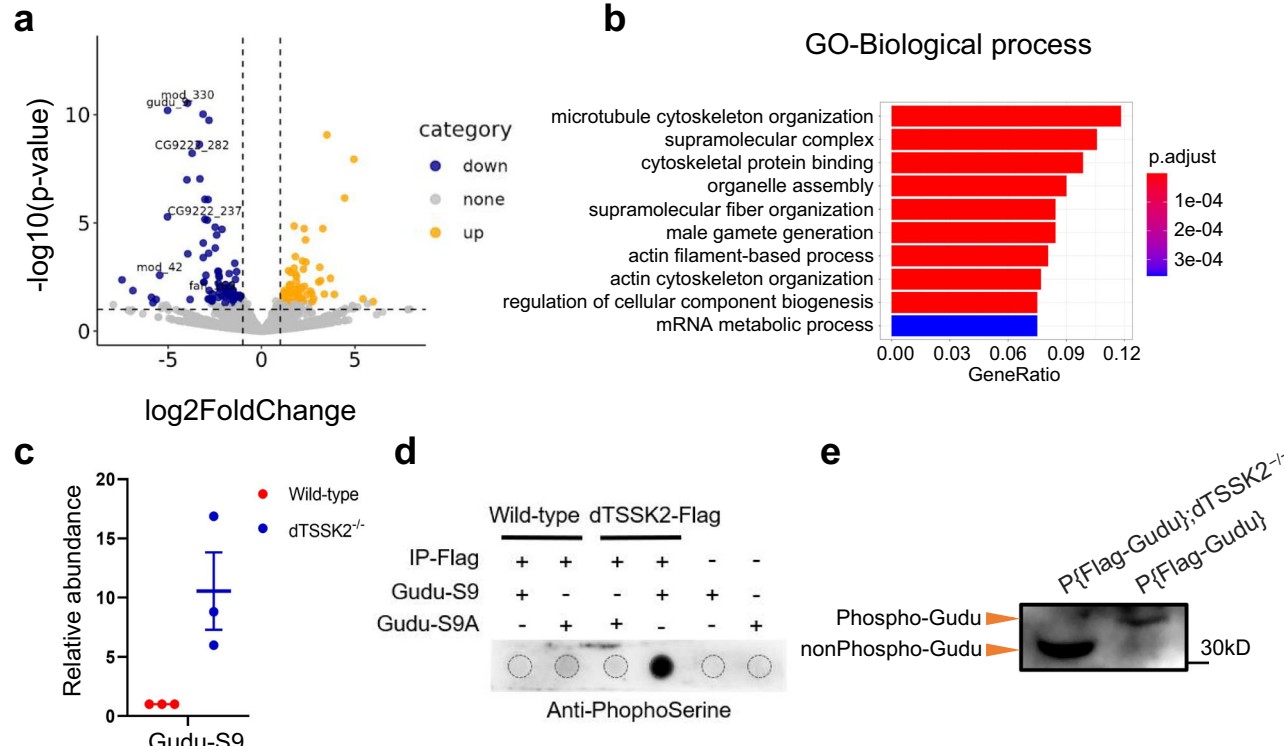

**Fig. 5 | Gudu is a potential substrate of dTSSK2. a** The volcano plot illustrates the differential abundance of potential substrates of dTSSK2. The plot's horizontal axis represents a *P* value of 0.1, and the vertical cut-off lines represent a log2 ratio of 1 (2 fold), which peptides were considered to exhibit significant abundance differences. Statistical analysis was conducted using a two-sided Student's *t* test. **b** Enrichment of Gene Ontology (GO) terms for phosphopeptides that are less abundant in dTSSK2$^{-/-}$ testes compared to wild-type testes. **c** Plots showing differences in the relative abundance of phosphopeptide of Gudu-Ser9 between wild-type (w$^{1118}$) and dTSSK2$^{-/-}$ flies (*n* = 3 per group). Data are mean ± SEM. Statistical analyses were performed by two-sided Student's *t* test. **d** Dot blot showing that purified Flag-dTSSK2 phosphorylates the peptide corresponding to Gudu-Ser9. A commercial antiphosphoserine antibody was used for dot blot detection. **e** WB analysis showing phosphorylation of dTSSK2 in testicular extracts of P{Flag-Gudu}; dTSSK2$^{-/-}$, P{Flag-Gudu} flies.

than its abundance. To directly test whether Gudu is a direct substrate of dTSSK2-mediated phosphorylation, we performed in vitro kinase assays using synthetic Gudu-Ser9 (wild-type) and Gudu-Ser9A (alanine-substituted) peptides. Dot blot hybridization assays with a serine phosphorylation-specific antibody confirmed that dTSSK2 phosphorylates the wild-type Gudu-Ser9 peptide, but not the alanine-substituted variant (Fig. 5d).

To validate this finding in vivo, we generated P{Flag-Gudu} transgenic flies in both wild-type (w$^{1118}$) and dTSSK2$^{-/-}$ genetic backgrounds. Phos-tag SDS-PAGE analysis of testis lysates revealed a distinct band shift corresponding to phosphorylated Gudu in wild-type flies, which was absent in the dTSSK2$^{-/-}$ genotype (Fig. 5e). These results strongly suggest that dTSSK2 directly mediates Gudu phosphorylation at Ser9 in the testes.

Next, we explored the functional significance of Gudu phosphorylation at Ser9. we generated Gudu$^{-/-}$ knockout flies, P{GuduS9A} transgenic flies in a Gudu$^{-/-}$ background, and P{Gudu} (wild-type) transgenic flies in Gudu$^{-/-}$ background. Fertility assays confirmed that Gudu$^{-/-}$ flies were sterile, as expected. By contrast, the Gudu$^{S9A}$ mutation in Gudu$^{-/-}$ did not significantly affect male fertility (Fig. S6a).

However, further examination of IC structure in the testes of Gudu$^{-/-}$; P{GuduS9A} flies revealed a partial disruption of the IC (Fig. S6b). Additionally, sperm motility assays indicated that the Gudu$^{S9A}$ mutation partially impaired sperm flagellar motility (Fig. S6c). These observations suggest that phosphorylation of Gudu at Ser9 by dTSSK2 contributes to proper IC function and optimal sperm motility, both of which are critical for male fertility.

To investigate the subcellular localization of Gudu, we performed in vivo imaging using GFP-tagged Gudu transgenic flies. Gudu was predominantly localized to the sperm flagella, consistent with its potential role in regulating motility. Importantly, the Gudu$^{S9A}$ mutation did not alter the

localization pattern of the protein (Fig. S6d), indicating that phosphorylation at Ser9 is not required for flagellar targeting.

Taken together, these findings establish Gudu as a physiological substrate of dTSSK2. The phosphorylation of Gudu at Ser9 by dTSSK2 partially influences IC integrity and sperm flagellar motility, underscoring its role in supporting spermiogenesis and male fertility in *Drosophila*.

## Discussion

In this study, we identified and characterized dTSSK2 (CG9222), a TSSK in *Drosophila melanogaster*, as a pivotal regulator of sperm flagellar motility and male fertility. Our findings expand the functional repertoire of the TSSK family and provide mechanistic insights into the evolutionary conservation of phosphorylation-dependent processes during spermiogenesis. Here, we discuss the broader implications of our findings, contextualizing them within established knowledge and highlighting their significance for both basic research and translational applications.

Our results demonstrate that dTSSK2 plays a conserved role in regulating sperm motility, as evidenced by the functional rescue of *Drosophila* dTSSK2 mutants by human TSSK4. This underscores the deep evolutionary conservation of the TSSK family. In mammals, TSSK members (TSSK1–TSSK6) exhibit distinct yet overlapping roles in sperm maturation. For example, TSSK4 knockout mice display flagellar structural defects and reduced fertility, phenotypes that mirror the motility impairments observed in *Drosophila* dTSSK2 mutants[11,14]. Our study bridges this evolutionary gap, demonstrating that *Drosophila* dTSSK2 shares functional homology with human TSSK4, particularly in phosphorylation-dependent motility regulation. The ability of human TSSK4 to rescue *Drosophila* mutants suggests the possibility of shared phosphorylation targets across species, such as cytoskeletal proteins or ion channels. These findings not only validate

*Drosophila* as a powerful model for studying human infertility mechanisms but also emphasize the therapeutic potential of targeting TSSK4 in male reproductive disorders.

Phosphorylation is a key post-translational modification (PTM) that dynamically regulates sperm motility by modulating cytoskeletal dynamics and ion channel activity[26,27]. In our study, phosphoproteomic analysis identified 143 differentially phosphorylated peptides in dTSSK2-deficient testes, many of which are associated with processes critical to spermiogenesis, such as microtubule organization, cytoskeletal remodeling, and male gamete generation. Among these, Gudu was identified as a direct substrate of dTSSK2. However, phosphorylation of Gudu at Ser9 resulted in only a partial improvement in sperm motility, suggesting that dTSSK2 regulates motility through a broader network of substrates.

This observation is consistent with studies in mammals, where TSSK-mediated phosphorylation of key proteins, such as ODF2 and TSKS, is crucial for flagellar assembly and function[14]. Interestingly, although the Gudu-S9A mutation moderately affected sperm motility and IC integrity, its limited phenotypic impact indicates that dTSSK2 function depends on additional substrates, suggesting potential functional redundancy. Similar redundancy has been observed in other *Drosophila* kinases, such as dTSSK (CG14305), which phosphorylates Mst77F and Mst33A during sperm chromatin remodeling[16]. It is likely that dTSSK2 operates in coordination with other kinases, such as Polo-Like Kinase (PLK), Mitogen-Activated Protein Kinase (MAPK), or dTSSK1, to ensure robust regulation of sperm motility. Future studies should aim to identify additional substrates and elucidate the broader phosphorylation network orchestrated by dTSSK2. This will provide further insight into the molecular mechanisms underlying sperm motility and fertility[28,29].

Sperm motility defects are a leading cause of male infertility, affecting ~20% of infertile men[30]. Our study provides a molecular basis for such defects by demonstrating that dTSSK2 governs flagellar beat frequency through phosphorylation-dependent mechanisms. In *Drosophila*, the loss of dTSSK2 results in severe motility impairments and disrupted sperm transit into the seminal vesicle, ultimately leading to male infertility. These findings are consistent with studies in humans, where mutations in TSSK4 or its substrates are associated with oligoasthenozoospermia, a condition characterized by reduced sperm concentration and motility[11,14]. Furthermore, Gudu homologs in mammals, such as SPATA16, have been implicated in male infertility, including globozoospermia[25]. These parallels suggest that dTSSK2-mediated phosphorylation pathways are clinically relevant. Targeting TSSK activity or substrate interactions could offer therapeutic avenues for addressing motility-related infertility in humans[8].

In addition to its role in sperm motility, dTSSK2 likely regulates other aspects of spermiogenesis. Our phosphoproteomic analysis revealed potential substrates involved in microtubule cytoskeleton organization (e.g., CCY), organelle assembly (e.g., Fan), and supramolecular fiber dynamics, suggesting that dTSSK2 coordinates multiple spermiogenic processes through phosphorylation-dependent pathways[9]. Interestingly, human TSSK4, which localizes to the sperm neck, has been proposed to regulate head-tail junction integrity, a conserved feature of sperm structure[20]. This raises the possibility that dTSSK2 may also influence other spermiogenic processes, such as acrosome formation or chromatin condensation. These processes are tightly regulated by other PTMs, such as ubiquitination and acetylation, which often act in concert with phosphorylation. Future research should explore whether dTSSK2 phosphorylation impacts these additional processes[31,32].

Our study integrates CRISPR/Cas9 mutagenesis, phosphoproteomics, and cross-species rescue experiments to elucidate the role of dTSSK2 in spermiogenesis. However, several limitations remain. First, while we confirmed Gudu phosphorylation by dTSSK2 in vitro, the structural basis of this interaction remains unclear. Structural studies, such as cryo-electron microscopy or molecular dynamics simulations, could provide deeper insights into substrate recognition and phosphorylation[21]; Second, while human TSSK4 rescued the dTSSK2 mutant phenotype, species-specific differences in substrate preference (e.g., ODF2 in mammals vs. Gudu in

flies) warrant further comparative studies[16]; Finally, the contribution of non-kinase regions in dTSSK2 function remains unexplored.

Male infertility remains a global health challenge, affecting ~5% of the male population[30]. Our study positions dTSSK2 and its substrates as potential diagnostic markers and therapeutic targets for motility-related infertility. For instance, small-molecule activators of TSSK4 could ameliorate motility defects, while anti-phospho-Gudu antibodies may serve as biomarkers for infertility diagnosis. Additionally, the *Drosophila* model provides a powerful platform for high-throughput screening of PTM-modulating compounds, accelerating drug discovery in reproductive medicine[8]. Future research should focus on expanding the substrate network of dTSSK2 and exploring its interplay with other PTMs. Furthermore, understanding the functional conservation of TSSKs in humans could pave the way for novel therapeutic approaches to male infertility and contraceptive development[33].

In conclusion, this study identifies dTSSK2 as a novel member of the TSSK family in *Drosophila*, with critical roles in regulating sperm motility and male fertility. By integrating genetic, biochemical, and phosphoproteomic approaches, we provide mechanistic insights into TSSK-mediated phosphorylation in spermiogenesis. These findings underscore the evolutionary conservation of TSSK function across species and highlight their potential relevance to human reproductive health.

## Methods

### *Drosophila* strains and maintenance
*Drosophila melanogaster* stocks were kept on cornmeal molasses food at room temperature (RT), and experimental crosses were raised at 25 °C. w[1118] flies were used for WT controls. For all experiments, males were used. The developmental stage or time of development at which analysis was done is indicated in the methods section. *Drosophila* strains used in this study and their origin are in Supplementary Table 3.

### Generation of transgenic lines
To generate transgenic flies expressing P{Flag-GFP-dTSSK2}, the endogenous promoter of CG9222 was applied from w[1118] genomic DNA using primers CG9222-pro-F/CG9222-pro-R. The amplied promoter sequence was cloned it into the pUAST-GFP-attB vector between the NotI and AgeI sites. The genomic region encompassing the CG9222 gene locus, including the start codon (ATG) and extending 1kb downstream of the coding region, was amplified with primers CG9222-F/CG9222-R and inserted into the pUAST-GFP-attB vector at the SpeI and KpnI sites. The resulting lasmids were then injected into attP2 embryos to achieve phiC31-mediated integration at the genomic attP landing sites. For the creation of P{dTSSK2} transgenic flies, the full CG9222 gene locus was amplified using primers CG9222-pro-F/CG9222-R and cloned into the pUAST-attB vector between the NotI and KpnI sites. To tagged proteins with fluorescent markers, the mCherry and GFP sequences were amplified from pHPdestmCherry (Addgene #24567) and pUAST-GFP-attB vectors using primers mCherry-F/mCherry-R and GFP-F/GFP-R, respectively. These amplified fragments were sequentially cloned into the pUAST-attB vector. For the generation of P{Mst35Bb-GFP} transgenic flies, the genomic region including the Mst35Bb promoter and gene body was amplified with primers Mst35Bb-F/Mst35Bb-R, while the 3′UTR was amplified with primers Mst35Bb-3′UTR-F/Mst35Bb-3′UTR-R. The GFP fragments were subsequently cloned into the preconstructed pUAST-attB vector to complete the construct. Similarly, to generate P{Flag-GFP-Gudu}, P{Gudu}, and P{Flag-Gudu} transgenic flies, the respective promoter, coding region, and 3′UTR were amplified from w[1118] genomic DNA and cloned into the pUAST-attB vector. To generate human TSSK4 transgenic fly lines, the endogenous promoter of CG9222 was amplified from w[1118] genomic DNA, and synthetic human TSSK4 sequences (GenScript) were cloned into the pUAST-attB vector. Point mutants of dTSSK2 and Gudu transgenic flies were generated by amplifying the respective gene coding region and cloning them into the T vector (Transgene #CT101-01). Site-specific point mutations were introduced using mutation-specific primers, and the resulting mutated

fragments were subcloned into the pUAST-attB vector. All plasmids were injected into embryos carrying the attP landing site for site-specific insertion using phiC31 integrase. The primers used for cloning and generating transgenic flies used in this study are provided in Supplementary Table 4.

## Generation of *Drosophila* mutants

CRISPR/Cas9-mediated mutagenesis was employed to generate dTSSK2[−/−] and Gudu[−/−] mutant fly lines, following a protocol previously established in our laboratory. Cas9 mRNA was transcribed in vitro utilizing the Sp6 mMESSAGE mMACHINE Kit (Thermo Fisher Scientific, #AM1340), and guide RNAs (gRNAs) were synthesized using the RiboMAX Large Scale RNA Production Systems-T7 Kit (Promega, #PR-P1320). Cas9 mRNA and gene-specific gRNAs were then purified and mixed to achieve final concentrations of 1 µg/µL and 50 ng/µL, respectively, prior to microinjection into w[1118] Drosophila embryos. The resulting mutants were subjected to PCR amplification and subsequent DNA sequencing to confirm the presence of the desired mutations. Primers used for gRNA construction, PCR amplification, and sequencing are listed in Supplementary Table 4.

## Sequence alignment and phylogenetic analysis

Protein sequences were aligned using ClustalW of Mega software[34]. Identity and similarity were calculated using Blastp[35]. The phylogenetic tree was constructed using Phylogenetic Analysis of Mega software[36].

## Fertility testing

Each male or virgin female from different mutant lines was mated with three w[1118] virgin females or males. The parental flies were maintained for a period of 7 days before being removed. Subsequently, the number of adult offspring from each cross was enumerated to assess the fertility of the mutant lines. For the qualitative fertility assay, males were grouped and tested in sets of ten. Individual virgin males were paired with a single w[1118] virgin female in separate vials, which were maintained at a constant temperature of 25 °C. Over the subsequent 15 days, the progeny resulting from each mating were transferred to fresh vials every 24 h. Once the progeny had fully emerged, the number of offspring from each vial was recorded. For each genotype combination, we calculated the mean number of offspring per parental pair along with the standard error of the mean. This analysis provided a quantitative measure of fertility for the different fly mutants under controlled conditions.

## Sperm tail-beat frequency analysis

To assess sperm motility, Sperm were extracted by dissecting the male seminal vesicle and releasing the contents into a 15 µl drop of PBS on a glass slide. Motility was assessed under brightfield conditions using a Zeiss Axio Imager Z2 with ApoTome 2. Raw video clips of 6 s were recorded from 4–6 distinct areas around the sperm mass using a ZEISS Axiocam 506 mono camera. Sperm tails exhibited varying beat frequencies, and we quantified the beat frequency of the 1–2 most rapidly beating tails per video clip to ensure consistent measurements.

For beat frequency analysis, recorded videos were processed in Fiji (RRID:SCR_002285) utilizing the ffmpeg plugin[37]. We measured the beat frequencies of the fastest-beating sperm tails, ensuring they were not overlapping or entangled with others. A selection line was applied to an isolated sperm tail section, and a 1-pixel "Multi Kymograph" was generated, plotting pixel intensities along the $X$-axis against frames on the $Y$-axis. The sperm tail beat appeared as a traveling waveform. We counted the number of beats and frames where the sperm tail was in focus and isolated. Beat frequency was calculated as follows: Hz = (no. of beats × 60 fps)/no. of frames.

## IF and microscopy

For comprehensive testis imaging, we selected testes from 3-day-old *Drosophila* males post-eclosion and carefully dissected them in phosphate-buffered saline (PBS). The testes were then transferred to a microscope slide with a small drop of PBS and covered with a coverslip for cytological examination. This method allowed for the visualization of the entire testis structure. For standard sperm imaging, we followed a similar protocol, dissecting 3-day-old testes in PBS and transferring them to a microscope slide with a coverslip to facilitate the release of testis contents. This approach provided a clear view of the sperm morphology. IC staining was conducted using TRITC-conjugated phalloidin, as previously described. Briefly, ten pairs of testes were dissected in cold PBS, incubated in sodium isocitrate for 5 min, and fixed in 4% paraformaldehyde (PFA) for 20 min. The tissue samples were flattened under a siliconized coverslip, frozen in liquid nitrogen, and then the coverslip was removed. The microscope slides were incubated in cold 1× PBST for 10 min, followed by RT 1× PBST for an additional 10 min, blocked in PBSTA (1× PBS containing 0.1% Triton X-100 and 5% bovine serum albumin), and incubated in 1× PBSTA containing TRITC phalloidin (Yeasen #40734ES75) for 1 h at 37 °C. After washing with 1× PBST, the samples were stained with DAPI. Confocal images were obtained using a Leica SP8 or Zeiss LSM 980 confocal microscope, and images were processed with Fiji software. Quantification of GFP fluorescence intensity in sperm was performed using ImageJ. We measured the area and integrated optical density within the sperm to calculate the mean signal intensity[38].

## SDS-PAGE, Phos-tag SDS-PAGE, and WB analysis

For total protein extraction, both transfected cells and fly tissues were lysed in 1.5× SDS loading buffer. Following brief sonication, samples were boiled at 95 °C for 10 min, centrifuged at 25,000 × $g$ for 20 min, and subjected to SDS-PAGE according to the manufacturer's instructions. Proteins were transferred onto a PVDF membrane at 25 V for 25 min using the Trans-Blot® Turbo™ Transfer System (Bio-Rad). Membranes were blocked with 5% BSA in PBSTween (PBS with 0.1% Tween 20) for 2 h at RT, then incubated with primary antibodies diluted in PBSTween containing 5% dry milk overnight at 4 °C. On the subsequent day, membranes were washed three times with PBSTween for 10 min each, incubated with secondary antibodies diluted in PBSTween for 1 h at RT with shaking, and washed again with PBSTween.

For Mn²⁺-Phos-tag SDS-PAGE, 40 µM MnCl2 and 20 µM Phos-tag acrylamide ligands were incorporated into a 12% separating gel prior to polymerization. Testicular samples were prepared in 1.5× SDS loading buffer, sonicated, boiled at 95 °C for 10 min, centrifuged at 25,000×g at 25 °C for 20 min, and electrophoresed at 80 V. Post-electrophoresis, gels were immersed in transfer buffer containing 10 mmol/L EDTA and agitated gently for 10 min, repeated three times. Gels were then equilibrated in EDTA-free transfer buffer for 10 min before proceeding with standard protocols. Western blot (WB) analysis was conducted using a rabbit polyclonal anti-Histone H3 antibody (Abcam, ab1791) at 1:1000 and a rabbit polyclonal anti-Flag antibody (Millipore, F7425) at 1:1000. Peroxidase-conjugated anti-mouse or anti-rabbit secondary antibodies (Yeasen, 33201ES60 or 33101ES60) were applied at 1:5000. WB signals were detected using a GE AI680UV instrument and imaged with Fiji software[38].

## Peptides

The peptides of Gudu-S9: "GTSSGTSHNRSRK" and Gudu-S9A: "GTSSGTAHNRSRK" are synthesized by GenScript.

## In vitro kinase assay

To evaluate the kinase activity of dTSSK2, we performed an in vitro kinase assay using Flag-tagged dTSSK2 immunoprecipitated from *Drosophila* testes. Approximately 100 pairs of testes of Flag-dTSSK2-expressing flies were dissected in cold PBS and centrifuged at 3000 × $g$ for 5 min at 4 °C. After discarding the supernatant, 600 µL of RIPA buffer (50 mM Tris-HCl, pH 7.4, 150 mM NaCl, 1% NP-40, 0.5% sodium deoxycholate, and 0.1% SDS) containing 1× protease inhibitor cocktail (Sigma), 1 mM PMSF, 1 mM NaF, 1 mM DTT, 0.5 mM Na₃VO₄, 0.5 mM EDTA, and 0.5 mM EGTA was added. The samples were briefly sonicated on ice and then centrifuged at 25,000 × $g$ for 30 min at 4 °C. The supernatant was incubated with Flag M2 gel (Sigma #A2220) for 5 h at 4 °C. After three washes with 1× Tris-buffered

saline (TBS), the beads-bound Flag-dTSSK2 was incubated with synthetic peptide substrates in kinase buffer (25 mM HEPES, 10 mM $MgCl_2$, and 0.5 mM EGTA) containing 1× protease inhibitor cocktail (Roche), 1 mM PMSF, 1 mM NaF, 1 mM DTT, 0.5 mM $Na_3VO_4$, 0.5 mM EDTA, and 0.5 mM EGTA for 30 min at 25 °C.

The reaction mixtures were then blotted onto a nitrocellulose membrane. The membranes were blocked by incubation with TBS containing 0.1% Tween-20 (0.1% TBSTween) and 5% skim milk for 2 h. Blots were washed with TBST and then incubated overnight at 4 °C with an anti-phosphoserine antibody (Merck #05-1000) was used at a 1:1000 dilution. After three washes with 0.1% TBSTween, the membranes were incubated with a peroxidase-conjugated secondary antibody for 1 h at RT. The membranes were washed again three times with 0.1% TBSTween and then treated with a chemiluminescent reagent. The chemiluminescent signals were detected with a GE AI680UV instrument, and the Images were processed with Fiji software.

### Quantitative phosphoproteomics and proteomics

On the seventh day post-emergence, ~200 pairs of testes were isolated from both dTSSK2$^{-/-}$ and wild-type *Drosophila* melanogaster and stored at −80 °C. For protein extraction, the frozen testes were lysed in 300 μl of lysis buffer containing 8 M urea and a phosphatase inhibitor cocktail (diluted 1:100). The lysate was subjected to ultrasonication (4 min; 3 s on; 13 s off) and then centrifuged at 14,000 rpm for 10 min at 4 °C to remove cellular debris. The protein supernatant was collected and was determined using the BCA assay (Pierce) with a multimode plate reader, and 400 μg of protein from each sample was taken for further analysis. Each sample was adjusted to a final volume of 200 μl with 50 mM $NH_4CO_3$. Add 5 mM DTT in the samples and incubate at 37 °C for one hour. Iodoacetamide (IAM) was then added to a final concentration of 10 mM, and the samples were vortexed and allowed to react in the dark at RT for 45 min. To reduce the urea concentration, 50 mM $NH_4CO_3$ was added to the protein solution to achieve a final concentration of 1 M urea. Trypsin (Promega) was added to the protein samples at a ratio of 1:50 (protein:enzyme) and incubated overnight at 37 °C. Add 1% trifluoroacetic acid (TFA) in the samples and adjust the pH to 2-4. The samples were then centrifuged at 14,000 rpm for 10 min at 4 °C, and the supernatant containing the peptides was collected. Desalting of the peptide samples was performed using an HLB column (Waters), and the eluted peptides were collected in 100 μl samples using the kit (Titansphere Phos-Ti0 Kit 200 μl 96pcs Export). The peptide fractions were then dried using a Speed-Vac (Thermo Scientific).

For LC-MS analysis, the dried peptides were reconstituted in 0.1% formic acid. Approximately 1 μg of the sample was subjected to proteomic analysis using an Easy-nLC 1000 system coupled with a Q Exactive HF (Thermo Scientific) mass spectrometer. The raw data were processed using the MaxQuant and Andromeda integrated search engine (v.1.5.4.1), and the statistical significance between dTSSK2 knockout and wild-type samples was assessed using a paired *t*-test. The remaining samples were primarily dedicated to phosphoproteomic studies. Peptide separation and analysis were performed using an Easy-nLC 1200 system coupled with an Orbitrap Fusion mass spectrometer (Thermo Scientific). Peptides, ~0.5 μg, were separated on a self-packed column (75 μm x 15 cm) filled with C18 AQ (5 μm, 300 Å, microm BioResources, Auburn, CA, USA) and eluted at a flow rate of 250 nL/min using a gradient of mobile phases A (0.1% formic acid) and B (0.1% formic acid + 80% acetonitrile). The gradient program included 50 min of 6–34% mobile phase B, followed by 3 min of 34-38%, 1 min of 38–90%, and a final 6 min at 90% mobile phase B, totaling a 60-min gradient. Full MS spectra (*m/z* range 350–1400) were acquired with a resolution of 120,000 at *m/z* 200 and a maximum ion accumulation time of 50 ms. Dynamic exclusion was set to 30 s. HCD MS/MS spectra were set at a resolution of 15,000 at *m/z* 200. The automatic gain control (AGC) targets were set to 5E5 for MS and 5E4 for MS2. For MS2, the isolation width was 1.6 *m/z* units, and the maximum ion accumulation time was 50 ms. Raw data processing and searching were conducted using MaxQuant software (v.1.6.5.0). Peptide identifications were filtered at a 1% false discovery rate

(FDR) at the peptide level, with a minimum peptide length of five amino acids. Intensities were normalized and subjected to downstream analysis. Normalized intensities were analyzed using Student's *t* test (two-tailed, assuming equal variance), and phosphopeptides with a *P* value < 0.05 and a fold change >1.5 were considered to have significantly different abundances between the samples.

### Statistics and reproducibility

Immunofluorescence and western blotting experiments were conducted with independently prepared samples on at least two separate occasions, yielding consistent results. Statistical analyses were performed using GraphPad Prism version 8.0, with methods detailed in the respective figure legends. Sample sizes were determined empirically rather than by statistical calculation.

### Reporting summary

Further information on research design is available in the Nature Portfolio Reporting Summary linked to this article.

### Data availability

The mass spectrometry proteomics data have been deposited to the iProX storage platform (http://www.iprox.cn/page/HMV006.html/) with the dataset identifier PX0010619000. The remaining data are available within the paper, Supplementary Information or Source Data file.

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

## Acknowledgements
This work was supported by grants from the National Natural Science Foundation of China (32070846 to G.G.). We thank the Molecular Imaging Core Facility (MICF) and Molecular and Cell Biology Core Facility (MCBCF) at the School of Life Science and Technology, ShanghaiTech University for providing technical support. We also thank the staff members of Mass Spectrometry team of ShanghaiTech University for technical support with the LC-MS/MS experiment.

## Author contributions
G.G. conceptualized and designed the overall project. P.J., S.A., J.Z., and Z.X. generated the fly mutants and transgenic flies. P. J. performed the fertility test and testis imaging. P.J. and S.A. performed the phosphoproteomic analysis. P.J., Z.X., and G.G. wrote the manuscript. Z.N. and S.A. carried out sequence alignments and protein structure predictions. All authors contributed to the discussion of the results and declared that there is no conflict of interest.

## Competing interests
The authors declare no competing interests.
