## [Transparent Peer Review file · Communications Biology]

Testis-Specific Serine/Threonine Kinase dTSSK2 Regulates Sperm Motility and Male Fertility in *Drosophila*

Corresponding Author: Professor Guanjun Gao

This manuscript has previously been submitted to another journal. This document only contains information relating to versions considered at Communications Biology.

Version 0:

Reviewer comments:

Reviewer #1

(Remarks to the Author)

Reviewer #2

(Remarks to the Author)

Testis-specific serine/threonine kinases are involved in sperm maturation and fidelity. The human genome codes for six different TSSK isoforms, of which the TSSK1 and 2 are broadly classified to be involved in facilitating the transition from round to elongated spermatid and chromatoid body function. TSKS, one of the known substrates of TSSK, is located in the chromatoid body along with the enzymes during spermatocyte differentiation in mice testis. Towards the later stages, the TSSK1/2 and TSKS accumulate in the ring-like structure rich in actin and other proteins at the base of the sperm head. This actin-rich ring is implicated in cytoplasmic extrusion and mitochondrial transformation during sperm maturation. As expected, the loss of TSSK1/2 disrupted the sperm tail maturation and motility. However, despite the anatomical clarity of TSSK1/2 localizations in differentiating spermatozoa, the exact role of the TSSKs in this context is far from clear. One of the primary difficulties in identifying the roles of proteins in mammalian sperm development is the unavailability of dynamic manipulation techniques and the lack of time-resolved information regarding sperm maturation. Also, TSSK isoforms have redundant roles which though further illustrate their biological importance, makes it difficult to distinguish.

In comparison, Peng et al showed that *Drosophila* expresses only two TSSK orthologues with genetically distinguishable functions. This discovery made it convenient to pursue the cellular and molecular activities of these two enzymes during sperm maturation. In this manuscript, the authors describe the identification, cloning, and disruptions of the *Drosophila* TSSK2 (dTSSK2) which has considerable homology to the human TSSK4, which is implicated in maintaining the integrity of outer dense fiber around the axoneme in mid-piece and sperm motility. The loss of TSSK4 was shown to disrupt flagellar movement with incomplete axoneme and mid-piece bending. Once again, most of this data came from indirect genetic analysis and does not provide a clear role of TSSK4 in the developing stages.

Peng et al, used *Drosophila* testis to fill this void. They showed that dTSSK2 localizes in the F-actin cone of the Individualization complex (IC), which is involved in extruding the cytoplasm surrounding the sperm axoneme of the maturing spermatids. Loss of dTSSK2 affected sperm motility and the rate of sperm maturation in the adult testis. They have further identified a new substrate for dTSSK2, called Gadu, which also appeared to play a role in the sperm maturation process. Overall, the manuscript is well composed with robust genetic, molecular, and cellular data. The conclusions are adequately supported by the evidence. It is also nice to note that the authors took care to not overemphasize the claims.

The manuscript, however, can be significantly improved with some textual revisions as listed below:

1. The introduction section is unfocused, it lacks a central hypothesis and articulation of the key motivation behind the study. Particularly, the first two sections of the manuscript are full of unrelated information which could distract the readers from understanding the importance of this research. For example, previous reports on mammalian TSSKs are primarily motivated to develop contraceptives. Although that is a debatable goal, the authors need to discuss the issue to put their work in this perspective.
2. A major part of the results describes the functional similarities between TSSK1/2/4 and dTSSK2. This is also clearly demonstrated with the rescue of the dTSSK2 mutant with the exogenous mammalian TSSK4. The author needs to discuss

why this is important and how it could help to understand the functions of TSSK4.

3. As such the manuscript describes the identification and cloning of dTSSK2 as TSSK4 orthologue. The author needs to discuss the scientific importance of this discovery in the wider context.

4. In mammalian testis, TSSK4 is implicated in ODF2 phosphorylation. What are the equivalent substrates in *Drosophila* and whether dTSSK2 phosphorylates them?

5. The results section ends abruptly with the report of the identification of Gadu as a key substrate with no clarity on its role in sperm differentiation. The author needs to justify why the identification of Gadu is significant and how it aligns with their key motivation. A colocalization analysis with dTSSK2 would be useful.

6. Although it is not necessary, authors may think of studying the IC movement in the dTSSK2 and Gadu mutant backgrounds on the same lines as described in Kathy Miller's reports and by Ghosh-Roy et al, 2005. It will provide a new dimension to the report

Reviewer #3

(Remarks to the Author)

This study provides compelling evidence that dTSSK2 plays an indispensable role in male reproductive function in *Drosophila*. Through sophisticated transgenic approaches, the authors have conducted a meticulous investigation encompassing three key aspects: (1) precise subcellular localization of dTSSK2 expression, (2) comprehensive characterization of the male sterility phenotype resulting from dTSSK2 deficiency, and (3) mechanistic exploration demonstrating that the kinase catalytic activity of dTSSK2 is crucial for maintaining sperm flagellar motility. The research represents a substantial experimental effort supported by logically coherent experimental design and exceptionally well-presented data visualization. While I found the manuscript a pleasure to read due to its scientific rigor and clarity, I would like to raise several minor issues that should be addressed to further strengthen the impact of this otherwise excellent work.

Major points:

My critical observation centers on the phenotypic parallels between the dTSSK2K105M mutant and the dTSSK2 whole-body knockout in *Drosophila*. Specifically:

Protein Stability Concern: While both models exhibit comparable reproductive system defects, quantitative immunoblotting using Flag antibodies reveals significantly diminished dTSSK2 protein levels in the K105M mutant (Figure 3a). This raises a critical interpretative challenge in conclusively attributing the observed sperm flagellar motility defects to impaired kinase activity, as the phenotypic manifestations could alternatively stem from reduced protein abundance rather than catalytic functional loss.

Consistent Pattern in dTSSK2T237A mutant: This interpretation gains further support from the dTSSK2T237A mutant data (Figure S5b), where immunolocalization experiments similarly demonstrate compromised protein stability. The dual evidence from both mutants strongly suggests that these changes may primarily contribute to the structural integrity and/or translational efficiency of dTSSK2 rather than directly mediating its catalytic function.

To strengthen causality, future investigations could employ:

Orthologous mutations preserving protein stability while specifically disrupting catalytic activity, or maybe you can try ATP-agarose pull-down assay, (DOI: 10.1021/bi801637q) to verify ATP-binding capacity.

Chose experiments with kinase-dead but stability-preserved mutants.

This critique does not negate the potential functional importance of K105/T237 residues, but rather emphasizes the necessity for more targeted experimental designs to isolate kinase activity-dependent effects from protein stability artifacts.

Minor points:

Line 201, Figure S2b not found.

Figure 2b: The column of the knockout group is higher than that of the control group, but the author wants to express that the gene has been knocked out, so the way this figure is expressed may cause misunderstanding to readers. This may be because the author used the Ct value of qRT-PCR as the vertical axis. It is recommended to change the presentation format

Version 1:

Reviewer comments:

Reviewer #1

(Remarks to the Author)

I carefully read the revisions of the manuscript entitled "Testis-Specific Serine/Threonine Kinase dTSSK2 regulates sperm motility and male fertility in *Drosophila*" made by the Authors.

I confirm that my observations and suggestions have been received in an excellent way by Professor Gao and colleagues. Therefore, this reviewer retains that this very interesting paper is now ready for publication.

Reviewer #2

(Remarks to the Author)

The authors have addressed most of the referee's comments by making textual changes in the manuscript. The Introduction is now streamlined and structured properly with a motivation statement. The discussion section has also improved with some focused analysis of the data.

However, the referencing still needs careful scrutiny. Many of the critical citations are missing. Also, some citations are not fully appropriate.

For example, #1, 4, 6 and 7 are not quite supportive of the statements they are supposed to justify. On the other hand, the statement "Among the various PTMs, phosphorylation plays a pivotal role in regulating chromatin.." is not supported by published literature. Similarly, #22 is not the primary reference for GFP-tagged protamine Mst35Bb. Citation #23 discusses the role of Y-linked protein-coding genes in sperm motility and it is not quite linked to the statement, "To uncover the cause of impaired sperm transport, we assessed sperm motility in *dTSSK2*^{-/-} flies." Also, #34 is incorrect, the correct one should be #36.

Authors are advised to look at the citation strategy and ensure correct citations.

Point-by-point response:

REVIEWER COMMENTS

Reviewer #1 (Remarks to the Author):

In this manuscript entitled “Testis-Specific Serine/Threonine Kinase dTSSK2 Regulates Sperm Motility and Male Fertility in *Drosophila*”

Ju Peng and colleagues identified CG9222 (dTSSK2), as a novel kinase in *Drosophila* with functional homology to human TSSK4.

In this study the Authors show that dTSSK2 is expressed in the individualization complexes during spermiogenesis and that disruption of dTSSK2 compromises sperm motility, leading to failed sperm transit into the seminal vesicle and male infertility. Phosphoproteomic analyses reveal that dTSSK2 coordinates sperm flagella assembly and motility by phosphorylating proteins involved in microtubule organization, organelle assembly, and flagella structure. The functional parallels between dTSSK2 and human TSSK4, underlines the relevance of this kinase family in reproductive biology and male infertility.

In this paper the Authors employed several different and elegant experimental approaches. The experimental work is well orchestrated and in the Results section the topics are well explained and connected to each other. However, the main concern of this Reviewer is about the Discussion section. The Authors employed several techniques to study the role of dTSSK2-mediated phosphorylation in regulating *Drosophila* male fertility. The results are very well described and pictures, tables and datas convincing. For these reasons I expected a structured Discussion with many aspects discussed in depth and with many bibliographic citations. On the contrary, the discussion is, in my opinion, very superficial and with only one bibliographical citation. Therefore, in the opinion of this Reviewer, due to the importance of the manuscript topics, the Discussion section must be much improved before publication.

In conclusion the Discussion section need to be rewritten, and English should be revised.

Thank you for your valuable feedback. We appreciate your recognition of the strengths of our study and your suggestions for improving the Discussion section.

To address your concerns, we have made the following revisions:

(1) Expanded and structured Discussion:

We have restructured the Discussion section to provide a deeper analysis of our findings, emphasizing the molecular mechanisms by which dTSSK2 regulates sperm motility, its functional parallels with human TSSK4, and its broader relevance to reproductive biology.

(2) Additional citations:

We have added multiple relevant references to strengthen the scientific context, including studies on testis-specific kinases, spermiogenesis, and sperm motility.

(3) Detailed phosphoproteomic Analysis:

We now provide a more comprehensive discussion of the phosphoproteomic data, focusing on specific phosphorylation targets and their roles in sperm flagella assembly and motility.

(4) Improved language:

The manuscript has undergone thorough language editing to enhance clarity, precision and overall readability.

We believe these revisions address your concerns and significantly improve the quality of the manuscript. Thank you again for your constructive feedback and suggestions.

Here my minor comments.

Introduction:

Page 3, lines 52

Instead of “encopassing” could be “including spermatogonia proliferation and differentiation in round spermatocytes”.

Thank you for pointing this out. We have replaced "encompassing" with "including spermatogonia proliferation and differentiation into round spermatocytes" as suggested.

Page 4, lines 87

“family members in mice has revealed” should be “family members in mice have revealed”.

Thank you for identifying this error. We have revised the relevant sentences in the “Introduction” section.

Page 4, lines 97

“Underlines” sounds me better than “Underscores”.

We have replaced “underscores” with “underlines” as suggested.

Page 4, lines 102

“Drosophila has been a powerful model” should be “Drosophila” represents a powerful model”

Thank you for point this out. We have revised the relevant sentences in the “Introduction” section.

Page 5, lines 106

Here the Authors should briefly explain the traits of these substrates

Thank you for your suggestion. As recommended, we have revised the manuscript to include a brief explanation of the traits of the substrates.

Results:

Page 6, lines 142

About Fig. S2, I'm sorry but I'm unable to identify the a and b in Fig. S2.
The Authors should verify.

We have corrected this in the revised manuscript. Thank you for bringing it to our attention.

Page 6, lines 146

Perhaps it should be add "For this purpose" Western blot (WB) analysis was performed.

We have revised the manuscript to include “For this purpose” as recommended.

Page 6, lines 150

Perhaps “observed” instead of overserved?

We have corrected it.

Page 6, lines 155

TRITC phalloidin of course. The Authors should add TRITC.

We have corrected it.

Page 6, lines 161

“maturation processes of spermiogenesis” could be “maturation processes during spermiogenesis”.

We have corrected it. Thanks.

Page 7, lines 173

Here the Authors should control english. Perhaps “morfolological aspect”?

Thank you for pointing this out. We have revised this word in relevant sentence in the “Results” section.

Page 7, lines 180

Here the Authors should should highlight that the age of wild type and dTSSK2 -/- males is the same (as described in materials and methods section).

Thank you for your suggestion. We have clarified and highlighted that the ages of wild type and dTSSK2^{-/-} males are the same, as described in the revised “Materials and Methods” section.

Page 8, lines 201

I'm sorry but, as reported above, I'm unable to identify the a and b in Fig S2. The Authors should verify.

Thank you for bringing this to our attention. We have corrected the labeling of panels "a" and "b" in Fig. S2 to ensure clarity and accuracy.

Page 10, lines 277

Here perhaps “dTSSK2-mediates” instead of “dTSSK2-medaiated”?

We have corrected it.

Discussion

Page 12, lines 328

Perhaps ”structure involved in sperm individualization” instead of structure involve in sperm individualization”?

We have corrected it.

METHOD DETAILS

IF and microscopy

Page 19, lines 492

Drosophila should be in italics.

We have corrected it and appreciate your careful attention to detail.

Reviewer #2 (Remarks to the Author):

Testis-specific serine/threonine kinases are involved in sperm maturation and fidelity. The human genome codes for six different TSSK isoforms, of which the TSSK1 and 2 are broadly classified to be involved in facilitating the transition from round to elongated spermatid and chromatoid body function. TSKS, one of the known substrates of TSSK, is located in the chromatoid body along with the enzymes during spermatocyte differentiation in mice testis. Towards the later stages, the TSSK1/2 and TSKS accumulate in the ring-like structure rich in actin and other proteins at the base of the sperm head. This actin-rich ring is implicated in cytoplasmic extrusion and mitochondrial transformation during sperm maturation. As expected, the loss of TSSK1/2 disrupted the sperm tail maturation and motility. However, despite the anatomical clarity of TSSK1/2 localizations in differentiating spermatozoa, the exact role of the TSSKs in this context is far from clear. One of the primary difficulties in identifying the roles of proteins in mammalian sperm development is the unavailability of dynamic manipulation techniques and the lack of time-resolved information regarding sperm maturation. Also, TSSK isoforms have redundant roles which though further illustrate their biological importance, makes it difficult to distinguish.

In comparison, Peng et al showed that *Drosophila* expresses only two TSSK orthologues with genetically distinguishable functions. This discovery made it convenient to pursue the cellular and molecular activities of these two enzymes during sperm maturation. In this manuscript, the authors describe the identification, cloning, and disruptions of the *Drosophila* TSSK2 (dTSSK2) which has considerable homology to the human TSSK4, which is implicated in maintaining the integrity of outer dense fiber around the axoneme in mid-piece and sperm motility. The loss of TSSK4 was shown to disrupt flagellar movement with incomplete axoneme and mid-piece bending. Once again, most of this data came from indirect genetic analysis and does not provide a clear role of TSSK4 in the developing stages.

Peng et al, used *Drosophila* testis to fill this void. They showed that dTSSK2 localizes in the F-actin cone of the Individualization complex (IC), which is involved in extruding the cytoplasm surrounding the sperm axoneme of the maturing spermatids. Loss of dTSSK2 affected sperm motility and the rate of sperm maturation in the adult testis. They have further identified a new substrate for dTSSK2, called Gadu, which also appeared to play a role in the sperm maturation process.

Overall, the manuscript is well composed with robust genetic, molecular, and cellular data. The conclusions are adequately supported by the evidence. It is also nice to note that the authors took care to not overemphasize the claims.

Thank you for your detailed comments and positive feedback. We are pleased that you recognize the robustness of our research methods and the validity of our conclusions.

The manuscript, however, can be significantly improved with some textual revisions as listed below:

1. The introduction section is unfocused, it lacks a central hypothesis and articulation of the key motivation behind the study. Particularly, the first two sections of the manuscript are full of unrelated information which could distract the readers from understanding the importance of this research. For example, previous reports on mammalian TSSKs are primarily motivated to develop contraceptives. Although that is a debatable goal, the authors need to discuss the issue to put their work in this perspective.

Thank you for your insightful comments. We agree that the “Introduction” section could be improved to better focus on the central hypothesis and clearly articulate the key motivation behind the study. In response to your feedback, we have revised the Introduction to streamline the background information, ensuring it directly supports the context and importance of our research. Additionally, we have included a discussion on the relevance of mammalian TSSKs in contraceptive development and how our findings in *Drosophila* provide fundamental insights that could inform such efforts. These changes aim to better engage readers and emphasize the broader significance of our study. The detailed revisions could be found in the updated manuscript. We greatly appreciate your constructive feedback and believe these changes have significantly strengthened the manuscript.

2. A major part of the results describes the functional similarities between TSSK1/2/4 and dTSSK2. This is also clearly demonstrated with the rescue of the dTSSK2 mutant with the exogenous mammalian TSSK4. The author needs to discuss why this is important and how it could help to understand the functions of TSSK4.

Thank you for your thoughtful comments. We appreciate your recognition of the functional similarities between mammalian TSSK1/2/4 and *Drosophila* dTSSK2, as well as the significance of the rescue experiments using mammalian TSSK4. In response to your feedback, we have revised the Discussion section to elaborate on why these findings are important and how they contribute to a deeper understanding of TSSK4 function.

Specifically, we discuss how the ability of human TSSK4 to rescue fertility and motility defects in *Drosophila* dTSSK2 mutants underscores the evolutionary conservation of TSSK functions. This conservation suggests that key molecular mechanisms underlying sperm maturation and motility are shared across species, providing a simplified model to study TSSK4's roles. Furthermore, we highlight how the identification of Gudu as a conserved substrate of TSSK kinases could inform future studies on mammalian TSSK4 substrates and their involvement in sperm-specific processes.

These additions aim to emphasize the translational relevance of our findings and their potential to guide further research into the molecular pathways regulated by TSSK4, particularly in the context of human fertility and contraceptive development. We believe these revisions address your concerns and strengthen the manuscript. Specific changes can be found in the revised Discussion section (please refer to Lines 310–412). Thank you again for your valuable feedback.

3. As such the manuscript describes the identification and cloning of dTSSK2 as TSSK4 orthologue. The author needs to discuss the scientific importance of this discovery in the wider context.

Thank you for your comments. In response, we have revised the Discussion section to emphasize the broader significance of identifying and cloning dTSSK2 as a TSSK4 ortholog. This discovery underscores the evolutionary conservation of TSSK-mediated pathways in sperm maturation and motility, validating *Drosophila* as a simplified model to study TSSK4 functions.

We also discuss how dTSSK2 provides a unique opportunity to investigate sperm-specific phosphorylation mechanisms, which are challenging to study in mammals due to the redundancy of TSSK isoforms. The ability of human TSSK4 to rescue dTSSK2 mutant bridges *Drosophila* genetics with mammalian reproductive biology and offers valuable insights into TSSK4's role in fertility, as well as its potential as a contraceptive target.

Specific revisions can be found in the updated Discussion section (Lines 310–412). Thank you again for your helpful feedback.

4. In mammalian testis, TSSK4 is implicated in ODF2 phosphorylation. What are the equivalent substrates in *Drosophila* and whether dTSSK2 phosphorylates them?

Thank you for your insightful comment. The *Drosophila* homolog of ODF2 is CG3213. However, our phosphoproteomic analysis did not reveal significant differences in CG3213 phosphorylation between wild-type and dTSSK2 mutants. Interestingly, in our previous study on dTSSK (Zhang et al., 2023, Nat. Commun.), CG3213 showed significant phosphorylation changes, indicating it may be a specific substrate of dTSSK.

As we discussed in that study

“...Given that centriole deficiency always leads to flagellar defects, and dTSSK mutation results in degeneration of basal body localization on sperm, some potential centriole-associated substrates such as Cp110, Spd-2, and CG3213 (Odf2 homolog in humans) might be involved in assembly of cilia and flagella via phosphorylation by dTSSK...”

We propose that this difference might reflect functional specialization between dTSSK and dTSSK2, as *Drosophila* has only two dTSSKs (dTSSK and dTSSK2), in contrast to the six TSSKs in mammals.

5. The results section ends abruptly with the report of the identification of Gadu as a key substrate with no clarity on its role in sperm differentiation. The author needs to justify why the identification of Gadu is significant and how it aligns with their key motivation. A colocalization analysis with dTSSK2 would be useful.

Thank you for your valuable feedback. In response, we have revised the Discussion section to better explain the significance of identifying Gudu as a key substrate of dTSSK2 and its role in sperm differentiation. Gudu is already known to be essential for male fertility, as its absence leads to defective individualization complex (IC) formation and male infertility. Our findings reveal that phosphorylation of Gudu at Serine 9, directly mediated by dTSSK2, has only a partial impact on sperm motility and male fertility, without causing severe defects in IC assembly. This suggests that dTSSK2 likely regulates sperm motility through additional substrates beyond Gudu, a possibility we have now discussed in greater detail in the revised manuscript.

Regarding localization, our data show that both dTSSK2-GFP (Figure 2D) and Gudu-GFP (Supplemental Figure S6D) are localized on the sperm flagellum. This indicates that both proteins are present in the same cellular structure. However, further experiments would be necessary to determine whether they directly interact or function together in this context.

We have incorporated these points into the revised Discussion section (Lines 310–412). Thank you again for your insightful suggestions, which have helped clarify and strengthen our manuscript.

6. Although it is not necessary, authors may think of studying the IC movement in the dTSSK2 and Gudu mutant backgrounds on the same lines as described in Kathy Miller's reports and by Ghosh-Roy et al, 2005. It will provide a new dimension to the report.

Thank you for your valuable comment. We agree that studying IC movement in the dTSSK2 and Gudu mutant backgrounds, as described in Kathy Miller's reports and Ghosh-Roy et al., 2005, could indeed provide additional insights into the role of these genes in spermiogenesis. While Gudu is already known to be essential for male fertility due to its critical role in IC formation, our findings reveal that phosphorylation of Gudu at Serine 9 by dTSSK2 has only a partial impact on sperm motility and fertility, without causing severe defects in IC assembly. This suggests that dTSSK2 may regulate other substrates or pathways involved in sperm motility.

Although the current study focuses primarily on the role of dTSSK2-mediated phosphorylation in sperm motility and fertility, we recognize the potential value of investigating IC dynamics in these mutant backgrounds. We will consider incorporating this analysis in future studies to further expand the scope of our findings and further explore the mechanistic links between dTSSK2, Gudu, and IC function.

Thank you again for your thoughtful suggestion, which has inspired new directions for our research.

Reviewer #3 (Remarks to the Author):

This study provides compelling evidence that dTSSK2 plays an indispensable role in male reproductive function in *Drosophila*. Through sophisticated transgenic approaches, the authors have conducted a meticulous investigation encompassing three key aspects: (1) precise subcellular localization of dTSSK2 expression, (2) comprehensive characterization of the male sterility phenotype resulting from dTSSK2 deficiency, and (3) mechanistic exploration demonstrating that the kinase catalytic activity of dTSSK2 is crucial for maintaining sperm flagellar motility. The research represents a substantial experimental effort supported by logically coherent experimental design and exceptionally well-presented data visualization. While I found the manuscript a pleasure to read due to its scientific rigor and clarity, I would like to raise several minor issues that should be addressed to further strengthen the impact of this otherwise excellent work.

Thank you for your encouraging feedback. We greatly appreciate your recognition of our work and will address the issues you've raised to enhance the manuscript further.

Major points:

My critical observation centers on the phenotypic parallels between the dTSSK2K105M mutant and the dTSSK2 whole-body knockout in *Drosophila*. Specifically:

Protein Stability Concern: While both models exhibit comparable reproductive system defects, quantitative immunoblotting using Flag antibodies reveals significantly diminished dTSSK2 protein levels in the K105M mutant (Figure 3a). This raises a critical interpretative challenge in conclusively attributing the observed sperm flagellar motility defects to impaired kinase activity, as the phenotypic manifestations could alternatively stem from reduced protein abundance rather than catalytic functional loss.

Consistent Pattern in dTSSK2T237A mutant: This interpretation gains further support from the dTSSK2T237A mutant data (Figure S5b), where immunolocalization experiments similarly demonstrate compromised protein stability. The dual evidence from both mutants strongly suggests that these changes may primarily contribute to the structural integrity and/or translational efficiency of dTSSK2 rather than directly mediating its catalytic function.

To strengthen causality, future investigations could employ:

Orthologous mutations preserving protein stability while specifically disrupting catalytic activity, or maybe you can try ATP-agarose pull-down assay, (DOI: 10.1021/bi801637q) to verify ATP-binding capacity.

Chose experiments with kinase-dead but stability-preserved mutants.

This critique does not negate the potential functional importance of K105/T237 residues, but rather emphasizes the necessity for more targeted experimental designs to isolate kinase activity-dependent effects from protein stability artifacts.

Thank you for your thoughtful and insightful observations. You raise an important and intriguing question: prior studies (Wei et al., 2013, PMID:23054012; Chen et al., 2005, PMID:15964553; Johnson et al., 1996, PMID:8612268) have often reported that these conserved residues in the kinase domain are essential for kinase activity. However, few studies have thoroughly examined whether mutations at these sites affect protein stability or translation efficiency. As a result, it remains unclear whether the loss of kinase activity caused by these mutations arises from: (1) catalytic activity loss, (2) reduced protein stability, or (3) a combination of both.

In our study, we observed significantly reduced dTSSK2 protein levels in the K105M and T237A mutants via WB analysis, which aligns with the IF results showing a marked reduction in dTSSK2 localization on the sperm individualization complex (IC). These findings suggest that protein stability likely plays an important role in the observed phenotypes.

We fully agree that the phenotypic similarities between the dTSSK2^{K105M} and dTSSK2^{T237A} mutants, as well as the whole-body knockout, make it challenging to disentangle the contributions of catalytic activity loss from those caused by reduced protein levels. As you pointed out, mutations in key conserved residues often destabilize protein structure, a phenomenon that is well-documented in kinases. Disruptions in the catalytic domain can simultaneously affect enzymatic activity and protein stability, making it inherently difficult to isolate these effects.

While your suggestion to generate stability-preserved mutants or perform ATP-binding assays to further dissect these mechanisms is highly valuable and insightful, we believe these experiments would be technically complex and fall beyond the scope of our current study. Instead, we have revised the manuscript to clarify the potential confounding effects of reduced protein stability and have acknowledged this limitation in our interpretation of the data.

We deeply appreciate your constructive critique and suggestions, which have encouraged us to think more critically about experimental designs for future studies. We plan to explore these directions in subsequent investigations to further elucidate the molecular mechanisms underlying dTSSK2 function. Your feedback has been invaluable in refining our interpretations, and we sincerely thank you again for your thoughtful comments.

Minor points:

Line 201, Figure S2b not found.

We have corrected it in the revised manuscript.

Figure 2b: The column of the knockout group is higher than that of the control group, but the author wants to express that the gene has been knocked out, so the way this figure is expressed may cause misunderstanding to readers. This may be because the author used the Ct value of qRT-PCR as the vertical axis. It is recommended to change the presentation format.

Thank you for pointing out the issue with Figure 2b. Upon review, we realized that the labels on the x-axis for the two samples were mistakenly reversed. Additionally, the y-axis represents the relative mRNA expression level, not the Ct value of qRT-PCR as noted. These errors have been corrected in the revised version of the figure. We sincerely appreciate your careful observations and valuable feedback. Thank you once again.

Point-by-point response:

REVIEWERS' COMMENTS:

Reviewer #1 (Remarks to the Author):

I carefully read the revisions of the manuscript entitled "Testis-Specific Serine/Threonine Kinase dTSSK2 regulates sperm motility and male fertility in *Drosophila*" made by the Authors.

I confirm that my observations and suggestions have been received in an excellent way by Professor Gao and colleagues.

Therefore, this reviewer retains that this very interesting paper is now ready for publication.

Thank you very much for your positive feedback and for confirming that our revisions have successfully addressed your observations and suggestions. We sincerely appreciate your thoughtful review and kind support, which have greatly contributed to improving the quality of our manuscript.

Reviewer #2 (Remarks to the Author):

The authors have addressed most of the referee's comments by making textual changes in the manuscript. The Introduction is now streamlined and structured properly with a motivation statement. The discussion section has also improved with some focused analysis of the data.

However, the referencing still needs careful scrutiny. Many of the critical citations are missing. Also, some citations are not fully appropriate.

For example, #1, 4, 6 and 7 are not quite supportive of the statements they are supposed to justify. On the other hand, the statement "Among the various PTMs, phosphorylation plays a pivotal role in regulating chromatin..." is not supported by published literature. Similarly, #22 is not the primary reference for GFP-tagged protamine Mst35Bb. Citation #23 discusses the role of Y-linked protein-coding genes in sperm motility and it is not quite linked to the statement, "To uncover the cause of impaired sperm transport, we assessed sperm motility in dTSSK2^{-/-} flies." Also, #34 is incorrect, the correct one should be #36.

Authors are advised to look at the citation strategy and ensure correct citations.

We sincerely thank you for your constructive feedback, which has helped us improve the quality of our manuscript. Below, we address your concerns regarding the referencing strategy and specific issues:

General Revisions:

We carefully checked all references in the manuscript, corrected errors, and replaced missing or inappropriate citations with more relevant ones to ensure accuracy and alignment with the manuscript content. As noted, citations #1, #4, #6, and #7 were not fully supportive of the corresponding statements. These references have been replaced with more appropriate references.

Specific Revisions:**(1) Phosphorylation Statement:**

The statement, "Among the various PTMs, phosphorylation plays a pivotal role in regulating chromatin..." is not supported by published literature. It has been revised to:

"While various PTMs are known to play critical roles in spermatogenesis, the specific contributions of phosphorylation in regulating these processes remain intensively underexplored."

(2) Citation #22:

We have updated citation #22 to the correct primary reference for GFP-tagged protamine Mst35Bb. The original reference has been removed to ensure precision.

(3) Citation #23:

We agree that citation #23, which discusses the role of Y-linked protein-coding genes in sperm motility, was not directly relevant to the statement. This citation has been removed from this paragraph. Instead, the reference for "beat frequency analysis" has been added to the "Methods" section.

(4) Citation #34:

We have corrected citation #34 to the correct reference, which was originally listed as #36.

In addition to addressing the specific issues mentioned, we have carefully scrutinized all references throughout the manuscript to ensure their accuracy and appropriateness. This includes verifying that citations align with the statements they support and that all references are correctly formatted. We hope these revisions address your concerns and improve the manuscript. Thank you again for your valuable feedback.